# Engineered Sleeping Beauty Transposon as Efficient System to Optimize Chimp Adenoviral Production

**DOI:** 10.3390/ijms23147538

**Published:** 2022-07-07

**Authors:** Samantha Baldassarri, Daniela Benati, Federica D’Alessio, Clarissa Patrizi, Eleonora Cattin, Michela Gentile, Angelo Raggioli, Alessandra Recchia

**Affiliations:** 1Centre for Regenerative Medicine, Department of Life Sciences, University of Modena and Reggio Emilia, 41121 Modena, Italy; samantha.baldassarri@gmail.com (S.B.); daniela.benati@unimore.it (D.B.); clarissa.patrizi2@gmail.com (C.P.); eleonora.cattin96@gmail.com (E.C.); 2ReiThera S.r.l., 00128 Rome, Italy; fed.dalessio@gmail.com (F.D.); michela.gentile@reithera.com (M.G.); angelo.raggioli@reithera.com (A.R.); 3Department of Molecular Medicine and Medical Biotechnologies, University of Naples “Federico II”, 80138 Naples, Italy

**Keywords:** Sleeping Beauty transposon, chimpanzee Adenovector production, precursor terminal protein

## Abstract

Sleeping Beauty (SB) is the first DNA transposon employed for efficient transposition in vertebrate cells, opening new applications for genetic engineering and gene therapies. A transposon-based gene delivery system holds the favourable features of non-viral vectors and an attractive safety profile. Here, we employed SB to engineer HEK293 cells for optimizing the production of a chimpanzee Adenovector (chAd) belonging to the Human Mastadenovirus C species. To date, chAd vectors are employed in several clinical settings for infectious diseases, last but not least COVID-19. A robust, efficient and quick viral vector production could advance the clinical application of chAd vectors. To this aim, we firstly swapped the hAd5 E1 with chAd-C E1 gene by using the CRISPR/Cas9 system. We demonstrated that in the absence of human Ad5 E1, chimp Ad-C E1 gene did not support HEK293 survival. To improve chAd-C vector production, we engineered HEK293 cells to stably express the chAd-C precursor terminal protein (ch.pTP), which plays a crucial role in chimpanzee Adenoviral DNA replication. The results indicate that exogenous ch.pTP expression significantly ameliorate the packaging and amplification of recombinant chAd-C vectors thus, the engineered HEK293ch.pTP cells could represent a superior packaging cell line for the production of these vectors.

## 1. Introduction

Human Adenoviral vectors, in particular those belonging to serotype 5 (Ad5), have been extensively used in clinical trials with an excellent safety profile, mainly as vectors for Human Immunodeficiency Virus (HIV) vaccines [1,2]. However, the broad tropism of human Adenovirus infections generates levels of host anti-vector immunity that may limit the utility of this vector [3]. Pre-existing immunity can be circumvented by employing vectors based on Adenoviruses isolated from different species, including chimpanzee Adenovirus (chAd) belonging to Human Mastadenovirus C species, which have significantly lower seroprevalence rates in human populations [4]. Indeed, while approximately 40% of humans have Ad5 neutralizing antibody (nAb) titers > 200, less than 10% of humans with nAb against chAd have titers higher than 200 [5,6]. Moreover, chAd showed comparable immunological potency to species C human Ad serotypes 5 and 6 in mice and primates [5] and can grow in human cell lines previously used for production of clinical material, such as HEK293 and PER.C6^®^ cells (Crucell, Leiden, The Netherlands). These features highlighted the advantage of chAd versus hAd5 as vaccine carrier in humans. To date, a number of clinical trials aimed at preventing the infection of Ebola, Malaria, HIV, Hepatitis C Virus, Respiratory Syncytial Virus [7] and recently SARS-CoV-2 are based on chAd vectors [8]. However, packaging and amplification of recombinant chAd in HEK293 cells remain a suboptimal time-consuming process. Thus, an engineered HEK293 cell line able to boost the chAd viral growth is needed.

A possible strategy to establish an engineered HEK293 cell line for chAd viral vector production exploits CRISPR-mediated genome editing. The CRISPR/Cas9 system is an effective genome editing tool relying on an engineered nuclease from *S. pyogenes* (SpCas9) driven by a guide RNA (gRNA) to the target DNA, where it induces double-strand breaks (DSBs) [9]. DSBs are repaired either by non-homologous end-joining (NHEJ) mechanism, an error-prone process leading to knockout of gene expression or by homology-directed repair (HDR) in presence of a DNA donor template [10,11]. The possibility to use the CRISPR/Cas9 system to induce specific genome editing events has been recently proposed as a novel therapeutic approach for monogenic disorders [12,13,14,15] and cancer [16], and could be employed to replace hAd5 E1 gene with E1 from chAd-C in HEK293 cells.

As alternative strategy, adenoviral factors could be modulated to improve vector production. Indeed, it has been shown that human Ad5 pre-Terminal Protein (pTP) overexpression enhanced viral replication of oncolytic Adenovirus in cancer cells [17]. Moreover, overexpression of human pTP in HEK293 cells was sufficient to significantly improve the rescue and propagation of recombinant human Ad5 vectors in the packaging cells [18,19]. No data on chimp pTP expression in HEK293 cells are reported so far. Thereby, here, we investigate whether the overexpression of chimp Ad-C pTP (ch.pTP) in HEK293 cells could benefit chimp adenoviral packaging. We exploited the highly efficient *Sleeping Beauty* system [20] to stably integrate the ch.pTP into the HEK293 genome. *Sleeping Beauty* (SB) belongs to the Tc1/mariner superfamily of DNA transposons [21] comprising a transposase gene flanked by inverted repeats (IRs) containing recognition sequences for transposase binding. A “cut and paste” mechanism mediates the transposition of SB in a new genomic locus. Sequence analysis of SB integration sites from mammalian cells showed that integration occurs exclusively into TA dinucleotides [22] and follows a fairly random distribution in the genome. This close-to-random integration profile of SB was confirmed by multiple studies and was reported for various organisms and cell types [23,24,25]. As a tool to integrate an exogenous DNA cargo in the human genome, SB was established 20 years ago [26] and is based on a two-component transposon system. During time, the transposase and the transposon DNA sequences have undergone stepwise optimization that resulted in the hyperactive SB100X transposase [27], the T2 transposon with fourfold increased transposition efficiency [28,29] and the “sandwich” (SA) transposon for large cargo DNA [24]. Thus, *Sleeping Beauty* could be delivered as a non-viral system through nucleofection [30] or lipofection of the two components, although several transposon-viral hybrid vectors based on Adenovirus [31], Adeno-associated virus (AAV) [32], Baculovirus [33] and integration defective lentiviral vector (IDLV) [25,34] have been developed.

To achieve our aim, we transposed [28] the ch.pTP expression cassette into HEK293 cells using SB system. The engineered HEK293ch.pTP bulk were compared to the parental HEK293 in rescuing and propagating chAd-C Venus vector [35]. Nine clones isolated from the HEK293ch.pTP bulk and expressing different level of ch.pTP were tested for rescuing and propagating the chAd-C Venus vector. 

Our results demonstrate that ch.pTP-overexpressing HEK293 cells improve yield and quality of recombinant chAd-C vectors. Moreover, clones #13 and #25 isolated from HEK293ch.pTP bulk represent good candidates for new packaging cells tailored to the production of chAd-C vectors, already employed in clinical trial for vaccine therapy.

## 2. Results

### 2.1. Successful Replacement of hAd5 E1 with chAd-C E1 Fails to Stably Immortalize HEK293 Cells

Several simian Adenovirus have been employed as E1-deleted Ad vectors for gene delivery or vaccine vectors exploited in human therapies [36,37,38,39]. Chimpanzee Ad vectors of E species belonging to Human Mastadenovirus E species (e.g., chAd63, ChAdOx-1) can be propagated on HEK293 cells, even if suboptimally [36]. However, vectors developed from chAd-C belonging to Human Mastadenovirus C species (Appendix A), are inefficiently propagated on regular HEK293 [40]. To optimize chAd-C vectors packaging into HEK293 cells, we firstly swapped hAd5 E1 with the chAd-C E1 gene, preserving the regulatory promoter/enhancer region of hAd5 E1 (Appendix A). To trigger HDR-mediated in situ replacement of the hAd5 E1, a chAd-C E1 expression cassette (donor chE1) and two gRNAs targeting the 5′-UTR sequence of hAd5 E1 just before E1A ATG (gRNA.E1A) and a region 20 nt-downstream E1B-55K STOP codon (gRNA.E1B) were generated. A single SpCas9 expression vector carrying the two gRNAs (pX330gRNA.E1A.gRNA.E1B) was co-transfected with the donor chE1 plasmid (pBS.chAd-C E1.HA) in HEK293 cells and the in situ HDR was followed over time by 3′ target integration (TI) PCR. As reported in Appendix A, HDR events successfully occurring from 2 days post-transfection (dpt), as confirmed by Sanger sequencing of PCR products (data not shown), were lost over time and completely disappeared 12 dpt indicating that HEK293 cells undergoing in situ replacement of hAd5 E1 with chAd-C E1 were negatively selected. To exclude the risk of poor transcription of chAd-C E1 from hAd5 E1 endogenous promoter, we constructed a donor template (pBS.PGK.chAd-C E1.HA) carrying the chAd-C E1 driven by the strong constitutive phosphoglycerate kinase promoter (PGK) (Appendix A). The expression of ch.E1 driven by endogenous or PGK promoter in HDR donor templates was confirmed by transfecting HeLa cells with donor ch.E1 (pBS.chAd-C E1.HA) and donor PGK.ch.E1 (pBS.PGK.chAd-C E1.HA) (Appendix A). Similarly, HDR-mediated in situ replacement occurred, however, HEK293 cells showing HDR events were lost over time (Appendix A). Collectively, these data confirm that in the absence of human Ad5 E1, chimp Ad-C E1 gene did not support HEK293 survival.

### 2.2. Sleeping Beauty System Mediates High Expression of ch.pTP in HEK293 Cells

To establish a stable HEK293 cell line able to boost the production and amplification of chAd vectors, we sought to permanently express the pTP from chAd-C (ch.pTP) belonging to Human Mastadenovirus C species (Appendix A and Appendix A) into HEK293 cells using the *Sleeping Beauty* system, which allows a genuine and safe integration profile [24] of the genetic cargo. The ch.pTP coding region driven by the hybrid CAGGS promoter, consisting of the cytomegalovirus (CMV) enhancer fused to the chicken beta-actin promoter and intron, was cloned into the pT2-based plasmid, generating the pT2.ch.pTP transposon. To determine the transposition efficiency, HEK293 cells were transfected with a size-comparable Venus-expressing T2 transposon alone or together with SB100X transposase. Venus expression, measured by flow cytometry all along the culture period (20 days), showed a signal that persisted in presence of SB100X and dropped without the transposase (Appendix A). Although the transposition rate was approximately 78.9% (Appendix A), we cloned a cassette for the expression of the Hygromycin resistance gene into the pT2.ch.pTP plasmid (pT2.ch.pTP.Hygro, Figure 1A) to facilitate the selection of cells expressing ch.pTP. Interestingly, HEK293 cells co-transfected with pT2.ch.pTP.Hygro and SB100X transposase coding plasmid (pCMVSB100X, Figure 1A) lost the ch.pTP in 8–15 days post-transfection in absence of antibiotic selection, as shown by PCR amplifying the CAGGS-pTP fragment (489 bp, Figure 1B), while a stable HEK293ch.pTP cell line was established by co-transfection of pT2.ch.pTP.Hygro and pCMVSB100X plasmids followed by Hygromycin selection. As control, HEK293 cells were transfected with pT2.ch.pTP.Hygro and selected with Hygromycin, according to the standard procedure to generate a stable cell line. The integration of ch.pTP expression cassette was followed over time (up to 26 dpt) by PCR on genomic DNA extracted from transfected cells, demonstrating the stable integration of ch.pTP in Hygromycin-resistant HEK293 cells independently of the transposase SB100X (Figure 1C). However, the expression of ch.pTP was higher in transposed HEK293 cells, as determined by semi-quantitative PCR, respect to Hygromycin-resistant non-transposed ch.pTP cells (Appendix A). Thereby, only transposed HEK293ch.pTP cells were used for further experiments. 

### 2.3. Stable Integration and Expression of ch.pTP Improves chAd-C Rescue and Propagation in HEK293 Cells

Firstly, we analysed whether ch.pTP expression in HEK293 cells enhances rescue and propagation of chAd-C Venus vector. According to a standard protocol for first-generation adenoviral vector [5], we transfected the *PmeI*-linearized plasmid DNA into engineered HEK293ch.pTP or parental HEK293 cells and measured the viral titers (viral particles per mL, VP/mL, and transducing unit per mL, TU/mL) rescued at P0 and propagated at passage 1 (P1). No further amplification steps were performed since the effect of the ch.pTP expression in the packaging cells would benefit preferentially the P0 of vector production which, in HEK293 cells, occurs in the absence of available initiator of Adenoviral replication. We measured, by Real Time PCR, the amount of ch.pTP mRNA produced during the packaging of a chAd-C vector at P0 from 3 dpt up to full cytopathic effect (15 dpt), in HEK293ch.pTP cells compared to HEK293 cells (Figure 2A). The analysis showed that the ch.pTP mRNA at 3 dpt was 2.5-fold higher than the transcript from the transgene integrated in the genome of HEK293ch.pTP cells, and increased up to 67-fold at 14 dpt. It is interesting to note that approximately 9 dpt, the level of ch.pTP mRNA produced during the packaging in HEK293 cells reached the basal level of ch.pTP mRNA in HEK293ch.pTP cells. The results of rescue and propagation experiments (*n* = 4) harvested at the same time frame (15 dpt) indicated that chAd-C Venus-rescue (P0) was significantly more efficient in engineered HEK293ch.pTP cells compared to the parental HEK293 (*n* = 3), as measured by VP/mL and TU/mL (Figure 2B,C). The propagation at P1 still gained a significant vector production compared to HEK293 cells as observed for TU/mL titer (Figure 2C) even though the physical titers (VP/mL) at P1 were comparable (Figure 2B). Correlating the amounts of physical particles to the transducing units, we looked at the quality of vector production, which is inversely proportional to the ratio between VP/mL and TU/mL: the lower the value of VP/TU, the higher the vector quality. The VP/TU ratio of the chAd-C vector produced in HEK293ch.pTP bulk was significantly lower compared to those produced in HEK293 cells at P0 (Figure 2D), indicating that ch.pTP expression during the rescue step (P0) also benefits the packaging of viral particles of significantly higher quality. To assess the stability of the bulk expressing ch.pTP, two rescue and propagation experiments for the chAd-C vector were performed in the HEK293ch.pTP bulks frozen 19, 30 and 48 days post-transfection and under Hygromycin selection. As reported in Appendix A, all HEK293ch.pTP bulks performed similarly in terms of VP/mL and TU/mL produced, indicating that the vector growth advantage provided by HEK293ch.pTP bulk persisted over time.

We challenged the HEK293ch.pTP bulk for the production of chAd-E EGFP vector belonging to the Human Mastadenovirus E species. The chAd-E EGFP-rescue (P0), measured by TU/mL, was approximately 4-fold more efficient in HEK293ch.pTP cells and the P1 in these cells produced 6-fold more transducing particles compared to HEK293 cells (Appendix A). Again, the quality of the vector packaged in the HEK293ch.pTP bulk was higher compared to the particles generated in the HEK293 cells (Appendix A). 

Once demonstrated the stable packaging advantages of the engineered HEK293ch.pTP cells, we performed cell cloning to isolate HEK293ch.pTP clones producing high-titer chAd-C vector. The expression of ch.pTP in 34 isolated clones was measured by RTqPCR (Figure 3A). Nine clones (#1, #2, #13, #17, #25, #26, #27, #30 and #36) with different levels of ch.pTP expression compared to HEK293ch.pTP cells were analysed for cell cycle progression (Figure 3B and Appendix A). The distribution of cells in G0-G1, S or G2/M phases did not show significant differences in HEK293ch.pTP bulk or clones, compared to HEK293 cells, suggesting that ch.pTP expression did not alter cell cycle progression. 

The selected nine clones were also tested, in two independent experiments, for the efficiency to package the chAd-C Venus vector compared to HEK293ch.pTP and HEK293 cells. Compared to HEK293 cells, all the selected clones showed a similar or higher production of chAd-C at P0 (Figure 4A,C), and a pronounced growth advantage at P1 (Figure 4B,D). Of note, clones #13 and #25 supported the chAd-C rescue and propagation at a high level, comparable to the HEK293ch.pTP cells as demonstrated by VP/mL and TU/mL titers at P0 (Figure 4A,B) and P1 (Figure 4C,D).

Overall, these data indicate that exogenous expression of ch.pTP can significantly improve the rescue and propagation of chAd-C vectors and new packaging clones for chAd-C vector production were isolated.

## 3. Discussion

The adenoviral E1-deleted vectors isolated from chimpanzees used so far, i.e., chAd-C show reduced yields upon propagation on HEK293 cells (10^2^–10^3^ VP/cell [41]) indicating that although they could grow in the presence of E1 from hAd5 virus, optimization of vector yield could be pursued.

Thereby, we replaced the human Ad5 E1 with chAd-C E1 in HEK293 cells with a CRISPR-based strategy. However, although chAd-C belongs to the highly immunogenic virus of C species, its E1 gene does not have the potential to stably transform human cells such as the hAd5 E1 gene. 

It is accepted that transformation of primary human cell lines with Adenovirus-derived sequences is a highly inefficient process, and it is possible that HEK293 cells have adapted to, and depend on the specific hAd5 E1 sequences present in their genome for survival. Indeed, in hAd5-E1-transformed human amniotic fluid cells (a mix of at least two different populations), only mesenchymal stem cell lines (but not cells of epithelial origin) could be derived, and in these cells the expression pattern of the different E1-derived proteins is different from the one observed in HEK293 cells [42], raising the possibility that interaction of hAd5 E1 DNA with specific host cell factors is crucial for proper expression of all E1 orfs.

Overall hAd5 and chAd-C E1 share 73.4% amino acid homology: 66.7% for E1A-13S, 65.8% for E1A-12S, 69.31% for E1B-19K and 82.64% for E1B-55K. It is possible that chAd E1-derived proteins interact less efficiently than their hAd5 counterpart with host cell factors necessary for transformation, including the Retinoblastoma protein (pRb), p53, and components of the DNA repair machinery like Mre11 [43]. This need for a homotypic interaction between E1 and its interaction partner would somehow be reminiscent of another growth property of Adenoviruses, i.e., vector developers using Adenoviruses other than hAd5 and growing them in cells harbouring the hAd5 E1 region (i.e., HEK293 or PER.C6) have historically replaced the native E4 orf6 with the corresponding one of hAd5 [44,45,46]. This vector modification is conducted to promote homotypic interaction between the E1 present in the packaging cell line and E4orf6 present on the vector, an interaction required to degrade p53 to improve viral yield. Interestingly, the ability of E1B-55K to bind and degrade its targets appears to be subgroup/serotype specific [47].

To pursue the aim of chAd-C growth optimization, we took advantage of the previously reported data claiming that overexpression of hAd5 pTP in HEK293 cells significantly improves the packaging and amplification of human recombinant Adenoviruses [18]. We employed the *Sleeping Beauty* system to stably integrate the ch.pTP in HEK293 cells. Although the transposition efficiency was remarkable, the hygromycin selection was required for the enrichment of cells expressing a level of ch.pTP compatible with HEK293 cell survival. Moreover, hygromycin selection without transposition events resulted in lower ch.pTP expression in the HEK293 bulk population, suggesting that a safe and genuine integration profile provided by the SB system benefits the expression of the ch.pTP.

The HEK293ch.pTP bulk and parental HEK293 were compared in the rescue and propagation of chAd-C Venus belonging to Human Mastadenovirus C species. HEK293ch.pTP cells significantly ameliorated the yield and the quality of chAd-C vector rescued at P0, underlining the key role of the pTP, the initiator of adenoviral replication. Indeed, the amount of chAd-C vector particles produced at P1 in HEK293ch.pTP and HEK293 cells were comparable, but the viral particles propagated in HEK293ch.pTP cells had a significantly higher infectious titer measured as TU/mL. Noteworthy, the ch.pTP integrated in the HEK293ch.pTP cell line shares over 99% homology at protein level with the pTP of the chimpanzee-derived Mastadenoviruses SAdV31.1, SAdV31.2, SAdV34, SAdV40.1 and SAdV40.2 (GenBank HC000816.1, HC000847.1, HC000785.1) reported by Roy et al. [48] hinting towards the possibility that the productivity of these and potentially of other Human Mastadenoviruses could also be improved by the use of this packaging cell line.

Not surprisingly, the chAd-E vector belonging Human Mastadenovirus E species benefits to a minor extent from HEK293ch.pTP cells, probably due to an uncomplete homology between the ch.pTP expressed in the HEK293ch.pTP and the pTP from chAd-E, which share an overall 79.05% identity.

During viral DNA replication, pTP interacts with two additional viral E2 gene products, namely the ssDNA binding protein DBP and the viral DNA polymerase, and two host cell factors, the nuclear factor I (NFI) and Oct-1. We reasoned that if ch.pTP in HEK293ch.pTP cells is able to promote viral growth of chAd-C and, to a lesser extent, of chAd-E, the problem might lie in the interaction with viral factors, rather than with cellular factors. The crucial amino acid for pTP interaction with Ad DNA polymerase in hAd5 is Ser580 [49]. This Ser lies within a conserved motif (D/E)XSX(D/E) also shared by the pTPs of chAd-C and chAd-E used in this study and is therefore unlikely to be accountable for the observed minor improved productivity of chAd-E versus chAd-C in the HEK293ch.pTP cells. Mutations in various additional residues of hAd5 pTP have been shown to impact its interaction with the Ad polymerase [50], but again, none of the mutations were present in chAd-E pTP. To our knowledge, fine mapping of the interaction between pTP and DBP is not available and therefore no speculation can be put forward on this. Nevertheless, it is noteworthy that pTP of chAd-E is considerably larger than the one of chAd-C (670 and 647 aa, respectively) and that this additional length is mostly due to insertions of 17, 4 and 7 consecutive amino acid stretches after residues 184, 365 and 387. 

The SB-mediated transposition of the ch.pTP in the HEK293 bulk confers a reproducible and stable production of chAd vectors over time allowing the isolation of new packaging clones for chAd vectors. Although no linear correlation between ch.pTP expression and chAd-C viral production was observed, clones #13 and #25 provided robust growth of the chAd-C Venus vector in two independent experiments. 

To conclude, we established a HEK293ch.pTP packaging cell line and we isolated new packaging clones to improve the generation and production of recombinant chAd-C vectors that would be beneficial for most applications in vaccine therapy.

## 4. Materials and Methods

### 4.1. Plasmids

To generate the effector plasmid pX330gRNA.E1A.gRNA.E1B, gRNA.E1A or gRNA.E1B were firstly cloned in the pX330-U6-Chimeric_BB-CBh-hSpCas9 plasmid, by oligo annealing in the *BbsI* site, and then the U6-gRNA.E1B cassette into the pX330.gRNA.E1A. For the HDR donor plasmid pBS.chAd-C E1.HA, the chAd-C E1 sequence from the last 46 nt of the 5′-UTR of E1A to 20 nt-downstream E1B STOP codon was amplified by PCR from pchAd-C E1. The promoterless chAd-C E1 sequence was cloned between the 5′ and 3′ homology arms (HA, 550 and 818 bp in length, respectively), amplified by PCR from hAd5 in HEK293 genomic DNA, and cloned in pBSIISK(−) plasmid. To generate the pBS.PGK.chAd-C E1.HA donor plasmid, the PGK sequence was cloned downstream of the 5′-HA, just before the chAd-C E1 sequence in the pBS.chAd-C E1.HA plasmid by digestion of the *NheI* site. The pCMVSB100X and pT2VENUS [24] plasmids were kindly provided by Prof. Zoltan Ivics, Paul Ehrlich Institute, Langen, Germany. To construct the pT2.ch.pTP plasmid, the ch.pTP coding region driven by the CAGGS promoter, followed by the bovine growth hormone (bGH) polyA site, was cloned between the T2 IR of the pT2/BH plasmid (provided by Prof. Ivics). To generate the pT2.ch.pTP.Hygro coding plasmid, the expression cassette for the Hygromycin resistance gene under the control of Herpes Simplex Virus Thymidine Kinase (TK) promoter was subcloned in pT2.ch.pTP plasmid downstream the polyA signal of ch.pTP expression cassette. 

### 4.2. Cell Culture, Transfection and Isolation of Single Cell Clones

HeLa and HEK293 cells were grown in cultured in Dulbecco’s modified Eagle’s medium (DMEM) supplemented with 10% foetal calf serum (FCS), 100 U/mL penicillin and 100 mg/mL streptomycin (Lonza Ltd., Basel, Switzerland). Transfection of 8 × 10^5^ cells was performed using Lipofectamine 2000 (Thermo Fisher Scientific, Monza, Italy) with 4 μg of total plasmid DNA. In transposition experiments, the transposon/transposase amounts of plasmid DNA was calculated accordingly to the stoichiometric ratio of 2:1 in a total quantity of 4 μg. In control samples, pCDNA3.1 plasmid (Thermo Fisher Scientific, Monza, Italy) was used instead of pCMVSB100X. The percentage of Venus^+^ cells was determined by flow cytometry at indicated time points post-transfection. The transposition efficiency was calculated as %Venus^+^ cells 20 dpt/%Venus^+^ cells 2 dpt × 100. For Hygromycin selection, starting from the day after transfection, cells were treated with 0.2 mg/mL of Hygromycin to select antibiotic-resistant cells. To isolate single-cell clones, 36 days post-transfection Hygromycin-resistant bulk were limiting-diluted to obtain a concentration of 0.3 cells/well and seeded in a 96-well plate.

### 4.3. Viral Vector Titration

The amount of viral particles produced during the rescue (P0) and propagation (P1), obtained by infecting HEK293s.pTP bulk or clones at MOI 150 (VP/mL), was measured using digital droplet PCR. In short, cells infected with the test vector were harvested at the indicated time point and subjected to three consecutive cycles of freeze/thawing at −80 °C/37 °C, followed by centrifugation at 2000 rpm at room temperature for 10 min to pellet debris.

Ten- or hundred-fold-diluted virus-containing supernatants were treated with DNAse (Roche, Merck Life Science, Milan, Italy) at 37 °C for 30 min to digest unpackaged DNA, according to the manufacturers’ protocol. Following EDTA treatment to inactivate the DNAse, DNAse-resistant viral genomes were released by incubation of the mix with SDS at final concentration of 0.1%, and following another ten- to hundred-fold dilution, DNA was subjected to digital droplet PCR-based quantitation using promoter-specific oligonucleotides (CMV Promoter Fw, CMV Promoter Rw, CMV Promoter Probe).

The infectious titers of stocks of chAd vectors were determined by endpoint dilution in 96-well plates containing 10^4^ HeLa cells per well. Forty-eight hours after infection with serially diluted chAd vector preparations, the percentage of EGFP or Venus positive cells was measured by acquiring on a FACS CANTO cytofluorimeter (BD Italy, Milan, Italy).

### 4.4. Analysis of CRISPR/Cas9-Mediated Editing on hAd5 E1 Gene and ch.pTP Integration in HEK293 Cells

Genomic DNA was extracted from HEK293 cells treated for CRISPR-mediated replacement or pTP transposition, using QIAamp DNA mini or micro kits (Qiagen, Milan, Italy) following the manufacturer’s instructions, at indicated time points post-transfection or transduction. To analyse the replacement of hAd5E1 with chAd-C E1, genomic DNA of HEK293 cells co-transfected with effector plasmid and HDR donor plasmid was amplified by PCR with primers ChAdE1-3HA-F2 and PSG4-RC (Appendix A) to analyse the correct integration of the donor at 3′ regions. The integration of ch.pTP expression cassette was determined by PCR on genomic DNA extracted from transfected HEK293 cells, with primers CAG-F and pTP-RC (Appendix A). As a control, the genomic DNA of transfected HEK293 cells was amplified by PCR with GAPDH-F and GAPDH-R (Appendix A).

### 4.5. Semi-Quantitative and Quantitative RT-PCR Analyses

Total RNA was isolated with the RNeasy Mini and Micro kit (Qiagen, Milan, Italy) according to the manufacturer’s instructions. cDNA was synthesized in a 20 μL reaction by Superscript III Reverse Transcription kit (Invitrogen, Thermo Fisher Scientific, Monza, Italy). To evaluate the expression of ch.pTP mRNA from the integrated transgene and during the viral packaging, SYBR green Real Time PCR was performed with ABI Prism 7900 Sequence Detection System (Applied Biosystems, Thermo Fisher Scientific, Monza, Italy) with Power SYBR Green PCR Master Mix (Thermo Fisher Scientific, Monza, Italy). Primers were designed to amplify both transcripts from the transgene and viral vector (pTP-SYBR-F and pTP-SYBR-R, Appendix A). Primers specific to human β-actin (SYBR-B-ACTIN-F and SYBR-B-ACTIN-R, Appendix A) were used as housekeeping control. Reactions were performed at 95 °C for 10 min, followed by 40 cycles at 95 °C for 15 s and 60 °C for 1 min, and dissociation curve analysis. To evaluate ch.pTP mRNA expression in HEK293ch.pTP bulks and clones, TaqMan Real Time PCR analysis was performed with ABI Prism 7900 Sequence Detection System (Applied Biosystems, Thermo Fisher Scientific, Monza, Italy) with TaqMan Universal PCR Master Mix. The ch.pTP expression levels were measured using custom assay (ch.pTP-F primer ch.pTP-R primer and ch.pTP-probe, Appendix A). Human *GAPDH* assay (hGAPDH: NM_02046.3, Applied Biosystem Thermo Fisher Scientific, Monza, Italy) was used as housekeeping control. Reactions were performed at 50 °C for 2 min and 95 °C for 10 min, followed by 40 cycles at 95 °C for 15 s and 60 °C for 1 min. The relative expression of the target genes was normalized to the level of *GAPDH* housekeeping gene in the same cDNA by using the 2^−ΔΔCT^ quantification. The replicated Relative Quantity (RQ) values for each biological sample were averaged. 

Semi-quantitative RT-PCR analysis to evaluate chAd-C E1 expression in HeLa cells transfected with ch.E1 donor plasmids was performed with ChAdE1-F and ChAdE1-R primers (Appendix A). Semi-quantitative RT-PCR analysis to evaluate ch.pTP expression in Hygromycin-selected bulks was performed with CAG-F and pTP-RC primers (Appendix A). As housekeeping control, GAPDH mRNA was amplified with GAPDH-F and GAPDH-R primers (Appendix A). PCR cycles: 95 °C 30 s, 58 °C 30 s, and 72 °C 30 s.

### 4.6. Cell Cycle Analysis

Cell cycle analysis was performed after 7 days of culture by flow cytometry according to Benatti et al. [51] Cell cycle phase distribution was determined using FlowJo software (Becton Dickinson, BD Italy, Milan, Italy).

### 4.7. Statistical Analysis

Statistics were computed with Prism version 9.0 software (GraphPad, San Diega CA, USA). Data were analysed for statistical significance using Student’s *t*-test. All values in each group were expressed as the mean ± SD. 

## Figures and Tables

**Figure 1 ijms-23-07538-f001:**
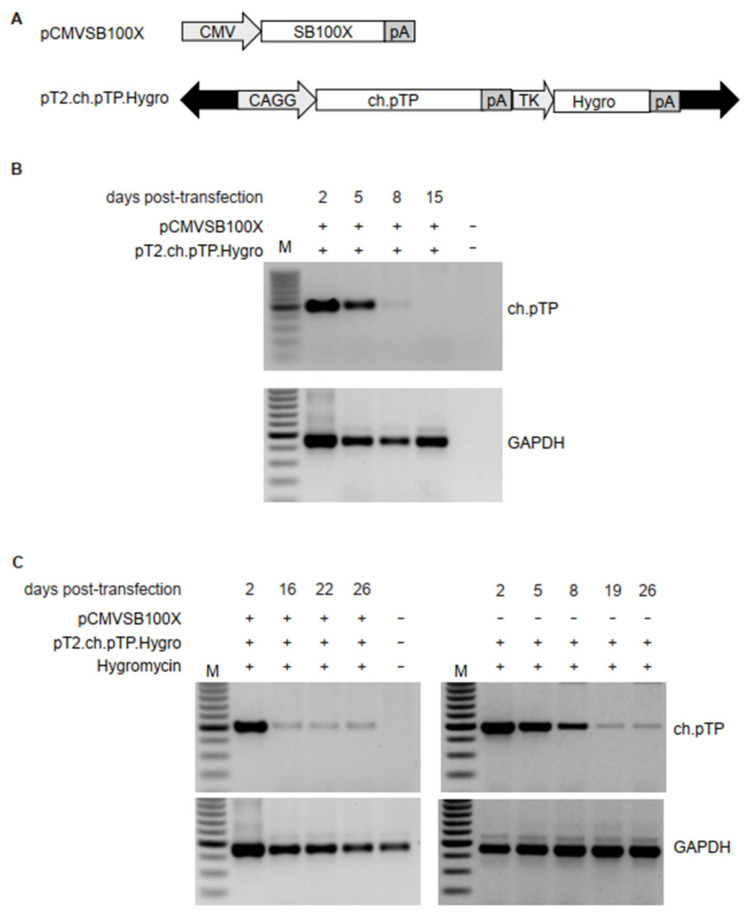
Transposition of ch.pTP in HEK293 cells. (**A**) Schematic outline of the plasmids carrying transposon or transposase. pCMVSB100X carries the SB100X transposase coding sequence under the control of cytomegalovirus (CMV) promoter. The pT2.ch.pTP.Hygro transposon plasmid carries the expression cassettes for ch.pTP, under the control of the hybrid CAGGS promoter, and a second expression cassette for the Hygromycin resistance gene (Hygro), driven by Herpes Simplex Virus Thymidine Kinase (TK) promoter. Black arrows indicate T2 IR. pA, polyadenylation signal. (**B**) PCR amplification of a fragment (489 bp) of ch.pTP and, as control, GAPDH in the genomic DNA from HEK293 cells co-transfected with pCMVSB100X and pT2.ch.pTP.Hygro plasmids. (**C**) PCR amplification of a fragment (489 bp) of ch.pTP and, as control, GAPDH in the genomic DNA from HEK293 cells transfected with pT2.ch.pTP.Hygro plasmid in presence (left panel) or absence (right panel) of pCMVSB100X and selected with Hygromycin. M, 100 bp molecular weight marker.

**Figure 2 ijms-23-07538-f002:**
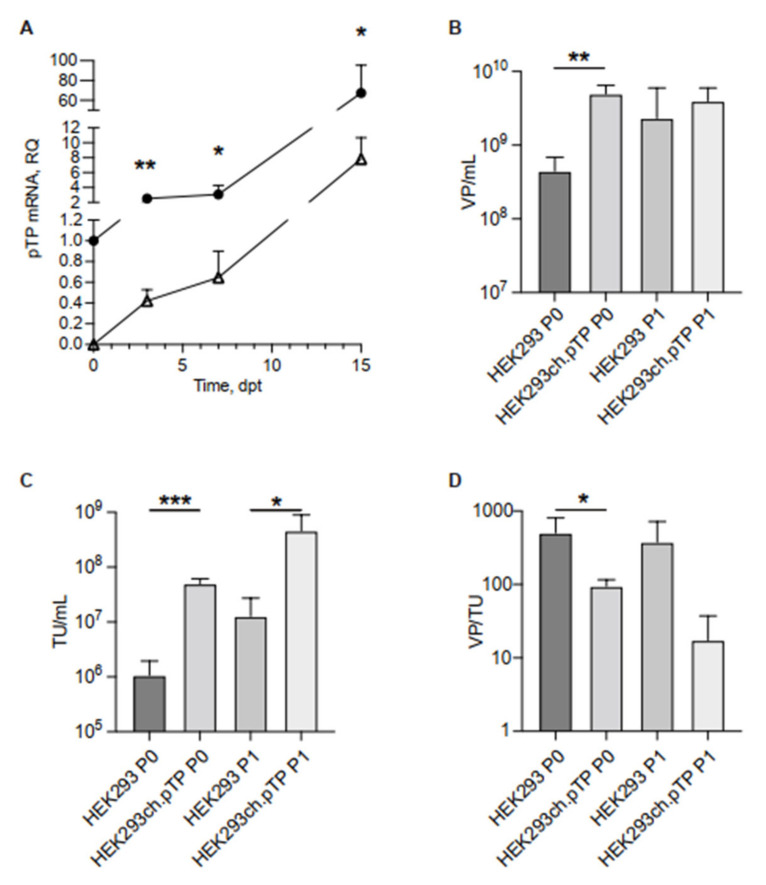
Transcript of ch.pTP and chAd-C vector produced in HEK293ch.pTP and HEK293 cells. (**A**) Real Time PCR analysis of ch.pTP transcript produced during the packaging of a chAd-C vector at P0 in HEK293ch.pTP cells (full circle) compared to HEK293 cells (empty triangle). The amount of ch.pTP mRNA was analysed from 3 dpt up to full cytopathic effect (15 dpt) in three independent experiments. (**B**) VP/mL and (**C**) TU/mL titration of chAd-C Venus vector rescued (P0) and propagated (P1) in HEK293 (*n* = 3) and in HEK293ch.pTP cells (*n* = 4). (**D**) Ratio VP/TU as a parameter of the quality of chAd-C vector rescued (P0) and propagated (P1) in HEK293 and HEK293ch.pTP cells. The data are reported as mean ± SD. *** *p* < 0.001; ** *p* ≤ 0.01; * *p* < 0.05.

**Figure 3 ijms-23-07538-f003:**
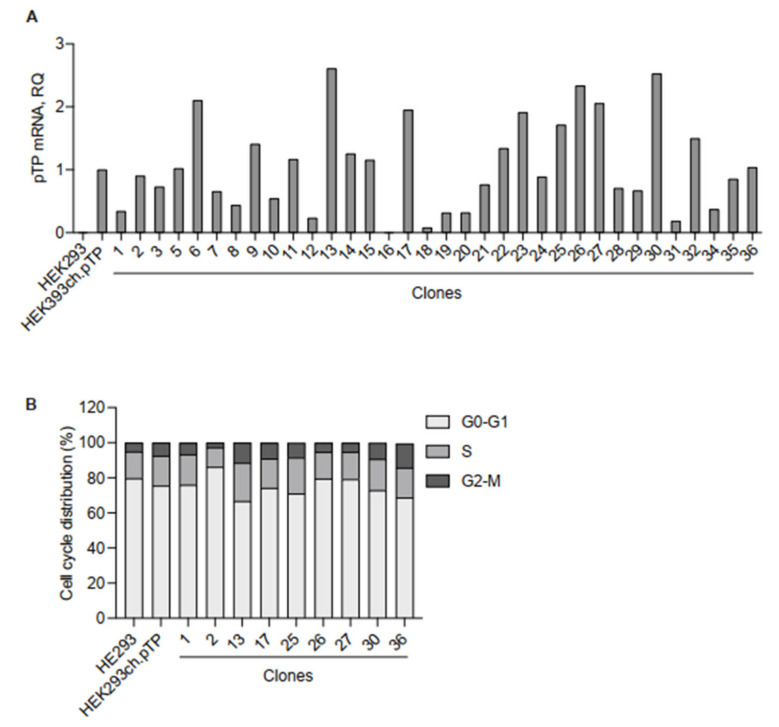
Molecular and cellular analysis of clones isolated from the HEK293ch.pTP bulk. (**A**) RTqPCR to assess the expression of ch.pTP in 34 clones compared to the HEK293ch.pTP bulk. As a negative control, ch.pTP expression in HEK293 cells was measured. The expression of *GAPDH* was used for normalization. (**B**) Cell cycle analysis by flow cytometry. The percentage of cells in each stage of the cell cycle (G0/G1, S, and G2/M) scored in HEK293, HEK293ch.pTP bulk and clones #1, #2, #13, #17, #25, #26, #27, #30 and #36 cultured for 7 days is indicated.

**Figure 4 ijms-23-07538-f004:**
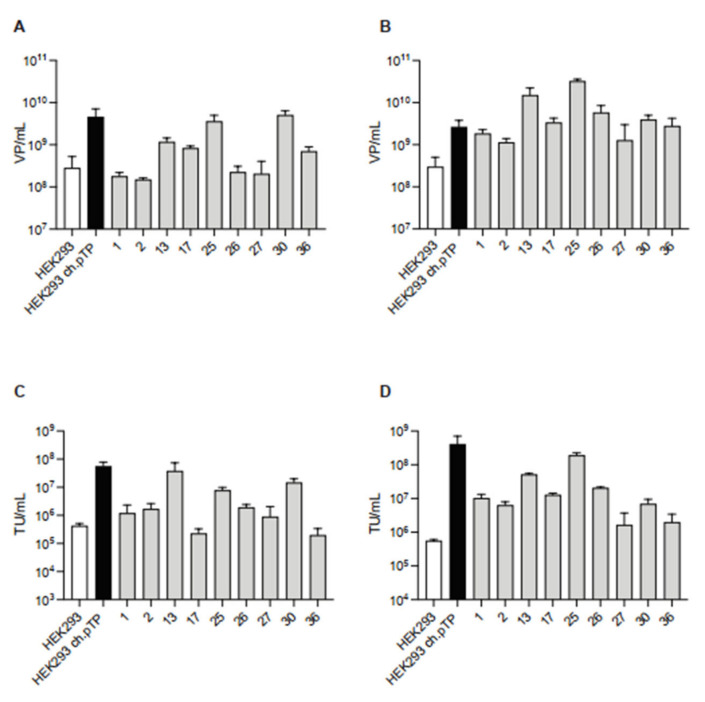
ChAd-C vector produced in HEK293ch.pTP clones. Titration of chAd-C Venus vector in terms of VP/mL and TU/mL rescued (P0, (**A**,**C**)) and propagated (P1, (**B**,**D**)) in HEK293, HEK293ch.pTP cells and nine isolated clones. Mean with standard deviation (*n* = 2) is presented.

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
