# Peer review of "Engineered Sleeping Beauty Transposon as Efficient System to Optimize Chimp Adenoviral Production"

_ijms, 2022, doi:10.3390/ijms23147538_

Round 1

Reviewer 1 Report

In their manuscript the authors report on several different approaches to improve packaging of recombinant chimpanzee adenoviral vectors (chAd) in HEK293 cells by supplying early chimpanzee adenoviral functions in this human cell line.

In their final experiments they use the sleeping beauty transposon (SB) system to establish HEK293 cell clones with stable expression of the chAd precursor terminal protein (ch.pTP). It is claimed that expression of this ch.TP significantly improves and accelerates the rescue and amplification of chAd vectors.

Of note, I have already reviewed the first submission of this manuscript to IJMS in April 2021. My rating was that the conclusion and claims of the authors “were only very poorly supported by their results“ (see attached pdf-file). Although the authors performed some formal changes in the re-submission, no really meaningful additional data was supplied.

The main difference in comparison to the former version of the manuscript consists of the selection of the HEK293ch.pTP cell clones shown in the detailed analysis (figures 3b and 4a/b of the current version and figure 4a/b and table 2 in the former version of the manuscript). However, the complete set of cell clones is obviously identical to that analyzed in the previous submission.

The conclusion that an improved packaging efficiency for the chAd-C venus vector correlates with expression level of ch.pTP is mainly based on clone #13 (lines 217 to 230).

For this clone #13, however, I noticed a strong discrepancy between the expression data in the first submission (fig. 4a/b, quite low levels as compared to the other clones) and in the resubmitted version of the manuscript (fig. 4a resubmission, highest level of all clones). Both analyses were performed by quantitative RT-PCR.

Independently of these discrepancies, a much larger selection of the generated cell clones should be subjected to the packaging analysis to be able to substantiate the claims.

Author Response

Reviewer 1

In their manuscript the authors report on several different approaches to improve packaging of recombinant chimpanzee adenoviral vectors (chAd) in HEK293 cells by supplying early chimpanzee adenoviral functions in this human cell line.

In their final experiments they use the sleeping beauty transposon (SB) system to establish HEK293 cell clones with stable expression of the chAd precursor terminal protein (ch.pTP). It is claimed that expression of this ch.TP significantly improves and accelerates the rescue and amplification of chAd vectors.

Of note, I have already reviewed the first submission of this manuscript to IJMS in April 2021. My rating was that the conclusion and claims of the authors “were only very poorly supported by their results“ (see attached pdf-file). Although the authors performed some formal changes in the re-submission, no really meaningful additional data was supplied.

Comparing the II and III revised manuscript to the former manuscript (April 2021) the authors significantly improved the manuscript. In particular, the authors:

  • performed 4 rescue and amplification experiments in HEK293ch.pTP bulk and 3 rescue and amplification experiments in HEK293, reporting a significant improvement of vector packaging in HEK293ch.pTP bulk.
  • measured vector titers as VP/mL and TU/mL demonstrating that the quality of vector particles produced in HEK293ch.pTP cells is superior (the quality of particles is a novel issue respect to the April 2021 submission)
  • screened 4 more clones in rescue and amplification of chAd-C Venus vector identifying 2 clones (#13 and #25) supporting high vector yield

The main difference in comparison to the former version of the manuscript consists of the selection of the HEK293ch.pTP cell clones shown in the detailed analysis (figures 3b and 4a/b of the current version and figure 4a/b and table 2 in the former version of the manuscript). However, the complete set of cell clones is obviously identical to that analyzed in the previous submission.

The panel of clones analyzed by Taqman RT PCR, and not by semiquantitative RT-PCR, is different from the former version of the manuscript (e.g., clone 3, 5, 6, 7 and 8 were not present in the first submission). Several clones were selected for rescue and amplification experiments, and cell cycle progression analysis was also performed in addition to the experiments of the former version of the manuscript (This analysis was not present in the former April 2021 submission).

The conclusion that an improved packaging efficiency for the chAd-C Venus vector correlates with expression level of ch.pTP is mainly based on clone #13 (lines 217 to 230).

The authors do not state that the expression of the pTP in the clone 13 linearly correlates with improved packaging efficiency. The sentence: “Clones #1, #2, #26 and #27 expressing lower level of ch.pTP compared to clones #13, confirmed the chAd-C growth advantage respect to HEK293 cells but were less efficient compared to the HEK293ch.pTP bulk”  was removed to avoid any misunderstanding.

The authors demonstrated that the HEK293ch.pTP bulk improved packaging efficiency of chAd-C vector and among 9 clones tested for vector production, clones #13 and #25 retrieved high vector yield (Figure 4)

For this clone #13, however, I noticed a strong discrepancy between the expression data in the first submission (fig. 4a/b, quite low levels as compared to the other clones) and in the resubmitted version of the manuscript (fig. 4a resubmission, highest level of all clones). Both analyses were performed by quantitative RT-PCR.

In the former version of the manuscript, the ch.pTP expression in clones isolated from HEK293ch.pTP bulk was measured by semi-quantitative RT-PCR, while in the revised version by quantitative Real Time Taqman RT-PCR.

Independently of these discrepancies, a much larger selection of the generated cell clones should be subjected to the packaging analysis to be able to substantiate the claims.

The authors do not aim to select and isolate the best clone able to sustain chAd-C vector packaging, neither to claim a linear correlation between ch.pTP expression and packaging efficiency. The authors aim to demonstrate that ch.pTP expression benefits the chAd-C vectors growth and that it is possible to isolate clones sustaining high vector yield.

Reviewer 2 Report

The authors describe that the generation and production of chimpanzee-derived adenoviruses can be improved by over expression of the chimp’s virus pTP gene. The overexpressing cell lines were generated using sleeping beauty modification. The manuscript is well written and easy to follow, but there are several aspects where the texts needs to be more detailed in both the taxonomy description of the virus used and the experimental conditions used to derive the conclusions. Such clarification is essential for grasping the conclusions of the manuscript.

  • Although the work is potentially interesting, the impact is strongly reduced by the confusing adenovirus terminology used in the manuscript. The ‘Chimpanzee adenovirus’ is not a valid virus taxon. It can be a member of for instance the mastadenovirus HAdV-C, HAdV-B or HAdV-E species, or of one of the simian mastadenoviruses (SAdV) species. The various species and their E1 proteins have markedly different biological effects in human cells. The reference to chAd-C is not particularly helpful as this is not a taxon according to the ICTV, and it could suggest relationship to either the SAdV-C or the HAdV-C species or subgroups.
  • The authors do not refer to the sequence of their adenovirus in for instance genbank. It would be very helpful to deposit the virus DNA sequence, and at least the E1 and pTP coding regions to allow the reader to inspect the diversity with the homologous sequences from HAdV-C5
  • Line 87 The sentence ‘The engineered HEK293ch.pTP bulk were compared to the parental HEK293 in rescuing and propagating chAd-C Venus belonging to C subgroup[35].’ is confusing. Ref 35 does not describe chimp adenoviruses and a transgene containing adenovirus itself should not be assigned to a species (of subgroup). Again is it the human or the simian subgroup (species) C?
  • How does the pTP mRNA quantity relate to the mRNA amounts seen in ‘chAd-C’ infection? This seems relevant info.
  • It is stated (line 249): Overall hAd5 and chAd-C E1 share 65% homology (66.7% for E1A-13S, 65.8% for E1A-12S, 69.31% for E1B-19K and 82.64% for E1B-55K. Is this based on amino-acid sequence alignments, or on the homology at the DNA level?
  • The authors replaced the Ad5 E1 sequences in 293 cells by the homologous sequences of their chAd-C. It is stated in line 106 that one of the guide RNAs targets a region 20 nt-downstream E1B STOP codon. Please confirm that this is the stop codon of the E1B 55k open reading frame.
  • In human HAdV-C viruses the first 2 or three codons of the pTP coding region are derived from an upstream exon. Are these included in the chAd-pTP construct? What is the sequence used of pTP gene?
  • What MOIs were used to measure virus yield in the assays to measure the P1 productions?
  • In line 279 a statement is made on the production of subgroup E viruses. It is not clear where these data come from, and which virus was used. The description ’ch.pTP from subgroup E’ is unclear as it could refer to HAdV-E or to SAdV-E. Also the precise virus should be mentioned as there could be differences between human and chimp viruses. If no data are shown, please indicate this by data not shown.

Author Response

Reviewer 2

The authors describe that the generation and production of chimpanzee-derived adenoviruses can be improved by over expression of the chimp’s virus pTP gene. The overexpressing cell lines were generated using sleeping beauty modification. The manuscript is well written and easy to follow, but there are several aspects where the texts needs to be more detailed in both the taxonomy description of the virus used and the experimental conditions used to derive the conclusions. Such clarification is essential for grasping the conclusions of the manuscript.

  • Although the work is potentially interesting, the impact is strongly reduced by the confusing adenovirus terminology used in the manuscript. The ‘Chimpanzee adenovirus’ is not a valid virus taxon. It can be a member of for instance the mastadenovirus HAdV-C, HAdV-B or HAdV-E species, or of one of the simian mastadenoviruses (SAdV) species. The various species and their E1 proteins have markedly different biological effects in human cells. The reference to chAd-C is not particularly helpful as this is not a taxon according to the ICTV, and it could suggest relationship to either the SAdV-C or the HAdV-C species or subgroups.

The reviewer is right. The chAd-C vector indicated in the former version belongs to simian mastadenovirus C species, and chAd-E to the Simian Mastadenovirus E species. This terminology was now stated in the introduction

The authors do not refer to the sequence of their adenovirus in for instance genbank. It would be very helpful to deposit the virus DNA sequence, and at least the E1 and pTP coding regions to allow the reader to inspect the diversity with the homologous sequences from HAdV-C5

The sequences of the Adenoviruses cannot be disclosed for intellectual properties in charge of ReiThera. Nevertheless, percentage of homology with HAdV-C5 are now reported in the main text.

  • Line 87 The sentence ‘The engineered HEK293ch.pTP bulk were compared to the parental HEK293 in rescuing and propagating chAd-C Venus belonging to C subgroup[35].’ is confusing. Ref 35 does not describe chimp adenoviruses and a transgene containing adenovirus itself should not be assigned to a species (of subgroup). Again is it the human or the simian subgroup (species) C?

The reviewer is right. The authors propagated the simian mastadenovirus C vector isolated from chimpanzee according to the classification described in ref. 35

  • How does the pTP mRNA quantity relate to the mRNA amounts seen in ‘chAd-C’ infection? This seems relevant info.

The authors measured the ch.pTP mRNA level provided by HEK293ch.pTP cells compared to the ch.pTP mRNA produced during the packaging of a chAd-C vector at P0 from 3dpt up to full cytopathic effect. The results are shown in Figure 2A.

  • It is stated (line 249): Overall hAd5 and chAd-C E1 share 65% homology (66.7% for E1A-13S, 65.8% for E1A-12S, 69.31% for E1B-19K and 82.64% for E1B-55K. Is this based on amino-acid sequence alignments, or on the homology at the DNA level?

Yes, it is. These percentages refer to amino acid sequences. This is now specified in the text

  • The authors replaced the Ad5 E1 sequences in 293 cells by the homologous sequences of their chAd-C. It is stated in line 106 that one of the guide RNAs targets a region 20 nt-downstream E1B STOP codon. Please confirm that this is the stop codon of the E1B 55k open reading frame.

The authors confirm. Now this detail is stated in the text

  • In human HAdV-C viruses the first 2 or three codons of the pTP coding region are derived from an upstream exon. Are these included in the chAd-pTP construct? What is the sequence used of pTP gene?

Yes, the first three codons are included in the ch.pTP construct. The ch.pTP sequence derived from chAd-C vector. No further details can be stated due to IP in charge of Reithera

  • What MOIs were used to measure virus yield in the assays to measure the P1 productions?

To perform the P1 the authors infected the packaging cells with vector lysate from P0 at MOI 150 (see materials and methods)

  • In line 279 a statement is made on the production of subgroup E viruses. It is not clear where these data come from, and which virus was used. The description ’ch.pTP from subgroup E’ is unclear as it could refer to HAdV-E or to SAdV-E. Also the precise virus should be mentioned as there could be differences between human and chimp viruses. If no data are shown, please indicate this by data not shown.

The reviewer is right. chAd-E belongs to simian mastadenovirus E species. Now this is indicated in the text.

Reviewer 3 Report

Adenovirus-based vectors have long been favoured for gene therapy applications, as well as for vaccine platforms, as exemplified by the successful use of the ChAdOx1-vaccine in fight against Sars-CoV2. Since chimpanzee adenoviruses (ChAd) have low seroprevalence in humans, pre-existing anti-vector immunity is not a concern for vectors based on these viruses. However, rescue, growth and propagation of ChAd vectors is still challenging due to a lack of suitable cell lines. The first generation E1/E3-deleted ChAd vectors grow rather inefficiently in HEK293, the cell line that contains the E1 region from human Ad-C5 and which is used for propagation of human subgroup C adenovirus vectors. In the current study, Baldassarri et al. engineered HEK293 cells for better rescue and growth of a ChAd-C vector. Their data show that CRISPR/Cas9-mediated replacement of the chromosomal human Ad-C5 E1 genes by the equivalent ChAd-C region did not improve ChAd-C vector production since chimpanzee Ad E1 genes did not support survival of the engineered HEK293 cells. Overexpression of human Ad-C5 pre-terminal protein (pTP; required for viral DNA replication) has been previously shown to improve rescue and yields of recombinant Ad-C5-based vectors in HEK293 cells, and, inspired by these results, the Authors tested also the effect of overexpression of chimp pTP on ChAd-C production. Stable, high level expression of the chimp pTP was achieved by Sleeping Beaty transposon-based transduction system, and the resulting HEK293ch.pTP cells improved infectious ChAd subgroup C vector yields by about 100-fold in comparison to HEK293 parental cells. No improvement in vector yields was observed for a ChAd subgroup E vector.  The Authors concluded that they have created a new chimp pTP-engineered HEK293 packaging cell line for improved generation and production of recombinant chimp Adenoviruses.

In general, the overall conclusion is supported by the data, but exactly how widely applicable their new packaging cell line is to chimp Adenovirus vectors is unclear since  the Authors did not disclose the precise viral origin of  their chimp pTP, or that of the chAd-C Venus vector used in the study. There are several type C chimp adenoviruses: are the pTP genes sufficiently conserved between these viruses for wide applicability of the newly created HEK293.ch.pTP cells?

Author Response

Reviewer 3

Adenovirus-based vectors have long been favoured for gene therapy applications, as well as for vaccine platforms, as exemplified by the successful use of the ChAdOx1-vaccine in fight against Sars-CoV2. Since chimpanzee adenoviruses (ChAd) have low seroprevalence in humans, pre-existing anti-vector immunity is not a concern for vectors based on these viruses. However, rescue, growth and propagation of ChAd vectors is still challenging due to a lack of suitable cell lines. The first generation E1/E3-deleted ChAd vectors grow rather inefficiently in HEK293, the cell line that contains the E1 region from human Ad-C5 and which is used for propagation of human subgroup C adenovirus vectors. In the current study, Baldassarri et al. engineered HEK293 cells for better rescue and growth of a ChAd-C vector. Their data show that CRISPR/Cas9-mediated replacement of the chromosomal human Ad-C5 E1 genes by the equivalent ChAd-C region did not improve ChAd-C vector production since chimpanzee Ad E1 genes did not support survival of the engineered HEK293 cells. Overexpression of human Ad-C5 pre-terminal protein (pTP; required for viral DNA replication) has been previously shown to improve rescue and yields of recombinant Ad-C5-based vectors in HEK293 cells, and, inspired by these results, the Authors tested also the effect of overexpression of chimp pTP on ChAd-C production. Stable, high level expression of the chimp pTP was achieved by Sleeping Beaty transposon-based transduction system, and the resulting HEK293ch.pTP cells improved infectious ChAd subgroup C vector yields by about 100-fold in comparison to HEK293 parental cells. No improvement in vector yields was observed for a ChAd subgroup E vector.  The Authors concluded that they have created a new chimp pTP-engineered HEK293 packaging cell line for improved generation and production of recombinant chimp Adenoviruses.

In general, the overall conclusion is supported by the data, but exactly how widely applicable their new packaging cell line is to chimp Adenovirus vectors is unclear since  the Authors did not disclose the precise viral origin of  their chimp pTP, or that of the chAd-C Venus vector used in the study. There are several type C chimp adenoviruses: are the pTP genes sufficiently conserved between these viruses for wide applicability of the newly created HEK293.ch.pTP cells?

The ch.pTP integrated in the HE293 cells derived from chAd-C virus and ameliorate the production of a first generation chAd-C Venus vector.

The point of the reviewer is well taken, and we now report in the main text that the pTP of the HEK293ch.pTP cell line described in this report shares over 99% homology at protein level with the pTP of the Chimpanzee-derived Simian Mastadenoviruses SAdV31.1, SAdV31.2, SAdV34, SAdV40.1 and SAdV40.2 reported by Roy et al (see references), hinting towards the possibility that the productivity of these and potentially of other simian mastadenoviruses could also be improved by the use of this packaging cell line. Due to the absence, in this report, of claims pinpointing the ameliorative effect of the pTP in the HEK293ch.pTP cell line to specific aminoacidic residues, any further level of homology with the pTP of additional Chimpanzee-derived Simian Mastadenoviruses group C can be calculated based on the one with the pTP of the SAdV described a few lines above.

Round 2

Reviewer 2 Report

Revised manuscript by Baldassarri and coworkers describes the optimization of the HEK293 cells for the production of chimpanzee-derived adenoviruses. The modified version clarifies some of the issues that were noted in the previous version of the manuscript.

However there are several issues remaining.

1.       The taxonomic status of the viruses used here is confusing and probably wrong. It is stated that the chimpanzee-derived virus belongs to the simian adenovirus C species. This is surprising. The other representants of this SAdV-C are derived from baboons, not chimps. Moreover, it is stated that: ’ Noteworthy, the ch.pTP integrated in the HEK293ch.pTP cell line shares over 99% homology at protein level with the pTP of the chimpanzee-derived simian mastadenoviruses SAdV31.1, SAdV31.2, SAdV34, SAdV40.1 and SAdV40.2 reported by Roy et al.[48] hinting towards the possibility that the productivity of these and potentially of other simian mastadenoviruses could also be improved by the use of this packaging cell line.’ The SAdVs listed are all members of the Human mastadenovirus C (HAdV-C) species, not SAdV-C. This level of similarity suggests that the chimp adenovirus used here may also be of the HAdV-C species.  The taxonomy needs to be clear.

2.       The same is true for data claimed to be obtained with the SAdV-E vector in lines 102-104. Again I doubt whether this is correct. The representative of this species SAdV-E was derived  from a velvet monkey, not a chimp. Probably the data refer to studies obtained with ape-derived viruses of species HAdV-E.

3.       Neither the sequences of the virus nor the sequences of the protein-coding regions are disclosed ‘for intellectual property reasons’. While such restrictions may be acceptable for minor disclosures that are not essential to the work, in this case it means that the work cannot be repeated by anyone.  So I would feel it essential to deposit either the entire DNA sequence of the virus or the sequences of the protein coding genes E1A, E1B and pTP of this virus. Without this info, the scientific value of the study is unfortunately rather limited.